# Excessive neutrophil recruitment promotes typical T-helper 17 responses in Coronavirus disease 2019 patients

Tanaka Arthur Choto[1,2]*, Ian Makupe[3], Andrew Zolani Cakana[4], Elopy Nimele Sibanda[3], Takafira Mduluza[1,2]*

1 Department of Biochemistry and Biotechnology, University of Zimbabwe, Harare, Zimbabwe, 2 Optics and Imaging, Doris Duke Medical Research Institute, College of Health Sciences, University of KwaZulu-Natal, Durban, KwaZulu-Natal, South Africa, 3 Gamma Allergy and Immunology Laboratory, Twin Palms Medical Centre, Harare, Zimbabwe, 4 The Haematology Centre, Harare, Zimbabwe

* taka.mduluza@gmail.com (TM); tchoto19@gmail.com (TAC)

## Abstract

Coronavirus disease 2019 (COVID-19) is caused by a recently identified virus, severe acute respiratory syndrome coronavirus 2 (SARS-CoV-2) and the disease is a pandemic. Although the hallmarks of severe COVID-19 have been established, the underlying mechanisms that promote severe pathology have not been thoroughly studied. A better understanding of the immune response in severe COVID-19 patients may help guide the development of therapeutic strategies and predict immuno-pathogenicity. This study was set to determine the lymphocyte and cytokine profiles associated with COVID-19 severity. A total of 43 hospitalised COVID-19 patients were recruited for the study and whole blood samples were drawn from each patient. Complete blood counts, lymphocyte subset profiles and C-reactive protein statuses of patients were determined. Cytometric bead array was performed to analyse the cytokine profiles of each patient. The demographic characteristics showed that the median age of the patients was 48.72 years, with an interquartile range from 40 to 60 years, and 69.77% of the patients were male. COVID-19 patients exhibited significantly low CD4+ lymphocyte expansion and leucocytosis augmented by elevated neutrophil and immature granulocytes. Stratification analysis revealed that reduced monocytes and elevated basophils and immature granulocytes are implicated in severe pathology. Additionally, cytokine results were noted to have significant incidences of interleukin 17A (IL-17A) expression associated with severe disease. Results from this study suggest that a systemic neutrophilic environment may preferentially skew CD4+ lymphocytes towards T-helper 17 and IL-17A promotion, thus, aggravating inflammation. Consequently, results from this study suggest broad activity immunomodulation and targeting neutrophils and blocking IL-17 production as therapeutic strategies against severe COVID-19.

**Data Availability Statement:** All data has been provided as within the manuscript and as "Supporting Information". However, any data underlying the results presented in the study are

available from the University of Zimbabwe College of Health Sciences Research Support Centre: Precious Chandiwana: precious. chandiwana@gmail.com and rsc@medschool.ac. zw http://uzchsrsc.ac.zw.

**Funding:** The study received partial funding from the TIBA to TIBA-Zimbabwe (TM & ENS). This research was commissioned by the National Institute of Health Research (NIHR), Global Health Research Programme (16/136/33) using UK aid from UK Government. The views expressed in this publication are those of the authors and not necessarily those of the NIHR or the Department of Health and Social Care. The funders had no role in study design, data collection and analysis, decision to publish, or preparation of the manuscript.

**Competing interests:** The authors have declared that no competing interests exist.

## Introduction

Coronavirus disease 2019 (COVID-19), is an infectious disease caused by a recently discovered coronavirus, severe acute respiratory syndrome coronavirus 2 (SARS-CoV-2), and the disease is a pandemic. The emergence and outbreak of the SARS-CoV-2 infections is considered to have occurred in December 2019, when pneumonia cases of unknown aetiology were identified in Wuhan, China [1]. Following the outbreak of COVID-19 cases in China, it did not take more than 4 months for COVID-19 cases to be vastly spread throughout the world. As a consequence, the spread of COVID-19 was declared a pandemic [2]. The COVID-19 pandemic presented an unprecedented burden to healthcare settings globally, with the progression of the pandemic being driven by successive waves of infection [3, 4].

COVID-19 is a notably heterogeneous disease according to clinical reports [5, 6]. Clinical presentations in COVID-19 patients can range from being an asymptomatic infection to critical illness that requires hospitalisation. At least 14% of infected patients show severe symptoms, often linked with imbalanced immune responses [7]. Critical COVID-19 cases are characterised by a cytokine storm syndrome and acute respiratory distress syndrome (ARDS), which may eventually lead to death [5, 6]. Dysregulated immune responses have been speculated to be the leading cause of morbidity and mortality [8]. A difficult task in the context of COVID-19 is providing comprehensive evidence of the underlying mechanisms that drive disease heterogeneity. Ideally, once enough evidence has been provided, the results may help guide the development of therapeutic strategies and predict immuno-pathogenicity [9, 10]. Akin to previous studies on SARS-CoV and MERS-CoV, patterns in the immune response and COVID-19 progression have a proximal association and may play a key role in disease severity [11, 12]. Hallmarks of severe COVID-19 cases have been widely described to be lymphopenia, aberrant granulocytes and monocytes, a cytokine storm and an increased neutrophil-to-lymphocyte ratio (NLR) [10, 13–15]. However, innate immune cells, particularly neutrophils, have been suggested to be the main mediators of immunopathology [16].

This current study was set to determine the lymphocyte subset and cytokine profiles associated with COVID-19 severity. COVID-19 severity in this case was determined by a NLR stratification and neutrophilia status stratification of hospitalised COVID-19 patients. Previous studies, systematic reviews and meta-analyses have demonstrated that the NLR can be utilised to diagnose and predict COVID-19 severity and outcome with remarkable accuracy [17–19]. More importantly, this approach significantly helps to ascertain the degree by which neutrophils modulate the immunopathology in COVID-19 patients.

## Materials and methods

### Study participants and clinical data

This study was granted ethical approval by the Medical Research Council of Zimbabwe (MRCZ/A/2602). The study was carried out in accordance to the principles and ethical guidelines of the International Declaration of Helsinki, the guidelines for Good Clinical Practice in Zimbabwe and the Medical Research Council Ethical Guidelines for Research. A total of 43 COVID-19 patients and 28 healthy individuals as controls were recruited for the study from Parirenyatwa General Hospital, Harare, Zimbabwe. Patient were recruited without a predetermined inclusion criteria (no inclusion with regards to age and sex), between 1 July 2020 and 30 November 2020. The patients' age and sex were recorded making use of the information collected as part of the hospital's procedures. All recruited participants were confirmed their COVID-19 positive status as part of the hospital's procedures, by detecting the presence of SARS-CoV-2 genetic material using reverse transcriptase polymerase chain reaction

(RT-PCR). Peripheral blood was drawn from each recruited patient on hospitalisation and the blood was collected into ethylenediaminetetraacetic acid (EDTA) vacutainers for subsequent assays.

## Determination of complete blood counts

A haematology analyser, Sysmex XN-3000™ (Sysmex Corporation, Kobe, Japan), was used to determine the complete blood counts of recruited patients within 2 hours of receiving whole blood specimens. The white blood cell differential (WDF) channel was used to determine differential white blood cell counts (lymphocytes, neutrophils, eosinophils and immature granulocytes). The global cell counts were determined by the white cell nucleated (WNR) channel. The complete blood counts were measured by aspirating approximately 88 μl of whole blood within 2 hours of reception. The haematology analyser then automatically determined the haematological parameters, making use of the radio frequencies (RF) and direct current (DC) method, hydrodynamic focusing, fluorescent flow cytometry and cyanide free sulfolyser method. The complete blood count results were recorded and captured for further analysis.

## Flow cytometric determination of lymphocyte subsets

The Becton Dickinson (BD) Multitest™ IMK kit, (Becton, Dickinson and Company BD Biosciences, San Jose, California, United States of America) was used to determine the lymphocytes subset populations. The 1X lysing solution was prepared by diluting the 10X concentrate of the BD Multitest™ IMK kit lysing solution with deionised water. Two 12 × 75 mm BD Trucount™ tubes were labelled with the sample identification number and the letters A and B to differentiate each tube. After verifying that the BD Trucount™ bead pellet was intact at the bottom of the BD Trucount™ tube, 20 μl of BD Multitest™ CD3/CD8/CD45/CD4 reagent was placed in tube A. Similarly, 20 μl of BD Multitest™ CD3/CD16+CD56/CD45/CD19 reagent was placed in tube B. In both cases, the reagent was pipetted to the bottom of the tube making sure to avoid the pellet. A volume of 50 μl of whole blood sample was placed at the bottom of both tubes. Both samples were stained by reverse pipetting, and were vortexed to ensure thorough mixing of the blood and the solution. After mixing, the tubes were incubated for 15 minutes in the dark at room temperature. A volume of 450 μl of 1X BD Multitest™ IMK kit lysing solution was added into both tubes and both tubes were vortexed to ensure thorough mixing. Subsequently, both tubes were incubated for another 15 minutes in a dark environment at room temperature. The samples in both tubes were sequentially analysed using a BD FACSCalibur™ flow cytometer (Becton, Dickinson and Company BD Biosciences, San Jose, California, United States of America), after incubation. BD Multiset™ software (Becton, Dickinson and Company BD Biosciences, San Jose, California, United States of America) was used to acquire and automatically measure the lymphocyte subsets of the samples. The procedure was repeated after the reception of each whole blood sample. The lymphocyte subset profile results were recorded and captured for further analysis.

## Qualitative analysis of serum C-reactive protein

Qualitative analysis of serum C-reactive protein was carried out using a C-reactive protein latex agglutination test kit (Fortress Diagnostics Limited, Antrim, Northern Ireland, United Kingdom). A volume of 50 μl of the serum sample and a drop of the positive control were placed on a card, in separate black circles. The latex reagent was re-suspended and a drop of the latex reagent was added to each black circle with the sample and positive control. The latex reagent was spread entirely over the area of the circle to ensure thorough mixing of the mixture. The cards were then rotated at 100 revolutions per minute for 2 minutes. Presence of

agglutination clumps similar to the positive control indicated a positive result (C-reactive protein of at least 6 mg/l). The process was done for all serum samples and the results were captured for analysis.

## Analysis of serum cytokine concentrations

The BD™ Human Th1/Th2/Th17 cytometric bead array (CBA) (Becton, Dickinson and Company BD Biosciences, San Jose, California, United States of America) kit was used to analyse cytokines in serum samples of COVID-19 patients. The kit allowed simultaneous detection of a set of cytokines (IL-2, IL-4, IL-6, IL-10, TNF-α, IFN-γ and IL-17A). The instrument used to detect the cytokines, the BD FACSCalibur™ flow cytometer (Becton, Dickinson and Company BD Biosciences, San Jose, California, United States of America), was setup according to the manual. The setup was done to configure the gating parameters of the instrument using the BD CellQuest Pro™ software (Becton, Dickinson and Company BD Biosciences, San Jose, California, United States of America). Lyophilised cytokine standards were reconstituted in 2 ml of the assay diluent for 15 minutes, and the tube was labelled as the top standard. After equilibration at room temperature, serial dilutions were made by transferring 300 μl of the top standard into the respective tubes containing 300 μl of assay diluent to create dilution ratios of 1:2, 1:4, 1:8, 1:16, 1:32, 1:64 1:128, and 1:256. Capture beads were then vortexed vigorously and the cocktail of capture beads the mixture was centrifuged at 200 *g* using a Hermle ZK364 centrifuge (Maschinenfabrik Berthold Hermle AG, Gosheim, Germany). After centrifuging, the capture beads were aspirated and re-suspended in serum enhancement buffer by adding the volume lost during aspiration. A volume of 50 μl of mixed capture beads was added to all tubes (50 μl aliquoted sample [COVID-19 hospitalised patients and healthy controls] and cytokine standards). The samples were incubated overnight in a dark environment to allow binding. The samples were then washed with 1 ml of wash buffer at 200 *g* for 5 minutes, after incubation. The pellet was then re-suspended in 300 μl wash buffer after centrifuging. The BD FACSCalibur™ flow cytometer (Becton, Dickinson and Company BD Biosciences, San Jose, California, United States of America) was used to acquire all samples and an acquisition template was used to record the results. The results were subsequently loaded for analysis on the FCAP Array™ application (version 3.0 for Windows® OS Becton, Dickinson and Company BD Biosciences, San Jose, California, United States of America). The software determined the mean fluorescence intensities (MFIs), which were fitted to a logistic curve-fitting equation to determine the concentrations of the cytokines. The determined concentrations of cytokines were captured for further analysis.

## Ethics statement

This study was conducted after the protocol was reviewed and approved by the Medical Research Council of Zimbabwe, approval MRCZ/A/2602. Permission to conduct the study was also obtained from the Joint Research Ethics Committee of Parirenyatwa Group of Hospitals and the University of Zimbabwe College of Health Sciences. After having thorough discussion on the procedures and purpose of the study and before commencement of data collection, written consent was obtained from the participants.

## Data analysis

After all the results were captured, statistical analyses and tests were done. Patients were stratified by their neutrophil to lymphocyte ratio (NLR) and neutrophilia status to determine the profiles that correlated with severe COVID-19. Calculation of the NLR was done by dividing the neutrophil absolute counts by the lymphocyte absolute count. Therefore, values above 7.5

were categorised in the high NLR group, which indicated severe immunopathology and values below 7.5 were categorised in the low NLR group, which indicated less severe immunopathology. The rationale behind using 7.5 as the cut-off value was to target neutrophilia patients (neutrophil count > 7.5 cells x $10^9$/L) and/or lymphopenia patients (lymphocyte count > 7.5 cells x $10^9$/L). These targeted patients are more likely to be severe COVID-19 cases. Therefore, a ratio of 7.5 and above was deemed as a suitable cut-off value. Patients were also stratified according by neutrophil counts, neutrophilia (> 7.5 cells x $10^9$/L) and non-neutrophilia patients (< 7.5 cells x $10^9$/L), in order to investigate how neutrophils exert their effects on lymphocyte subset expansion. The Mann-Whitney $U$ test and Kruskal-Wallis test were used to compare the distributions. Spearman's correlation and Fischer's exact tests were used to investigate association between variables. All data was analysed using statistical software, STATA (version 16.0, StataCorp Limited Liability Company, Texas, USA) and Graphpad Prism 5® (Version 5.0, Graph pad Software Inc, San Diego, United States of America). Results with $p$-values less than 0.05 (< 0.05), were statistically significant.

## Results

### Demographic characteristics of recruited COVID-19 patients

The aim of this study was to determine the lymphocyte and cytokine profiles that are associated with COVID-19 severity. Therefore, a total of 43 hospitalised COVID-19 patients were recruited for the study from Parirenyatwa General Hospital in Harare. Demographic analysis showed that 13 (30.23%) of the patients were female and 30 (69.77%) of the patients male. Additionally the median age of the collective group of patients was 48.72 years and the interquartile range was 40 years to 60 years (Table 1).

### Summary of haematological features and lymphocyte profiles

After complete blood counts and lymphocyte profiles were measured, the results summarised in Tables 2 and 3 respectively. The results were and recorded together with their reference ranges. The reference ranges were used as guidelines to infer anomalies caused by COVID-19. Therefore, COVID-19 patients were characterised by CD4$^+$ T-cell lymphopenia and their white blood cell differentials were skewed towards higher neutrophil and immature granulocyte percentages and lower lymphocyte percentages.

### Spearman's correlation analysis of leucocyte subsets

A two-tailed Spearman's correlation test was carried out at 95% confidence interval, to investigate association between leucocyte subsets. The results of the analysis were displayed in a correlation matrix (Fig 1) and a $p$-value table (Table 4). Lymphocyte counts showed the strongest positive correlation with monocyte counts ($r_s = 0.63$; $p < 0.0001$). Basophil demonstrated significant correlations with all leucocyte subsets. Basophils were positively correlated with

**Table 1. Demographic characteristics of the patients recruited for the study.**

| Variable | Descriptive Statistic of Recruited COVID-19 Patients |
|---|---|
| $n$ | 43 |
| Age in years, median (Q1, Q3) | 48.72 (40, 60) |
| Sex: | |
| Female, $n$ (%) | 13 (30.23) |
| Male, $n$ (%) | 30 (69.77) |

**Table 2. Summary of haematological characteristics of hospitalised COVID-19 patients.**

| Variable | Hospitalised COVID-19 Patients Median (Q1, Q3) | Reference Range |
|---|---|---|
| White Blood Cell Count (x $10^9$/L) | 10.045 (7.665, 12.585) | 4.5–11 |
| Red Blood Cell Count (x $10^{12}$/L) | 4.43 (3.385, 4.93) | 4.65–6.5 |
| Haemoglobin Count (g/dl) | 12.46 (10.25, 14.15) | 13–18 |
| Haematocrit (%) | 43 (33.75, 47.7) | 43–55 |
| Mean Corpuscular Volume (fl) | 93.95 (88.6, 103.7) | 77–95 |
| Mean Corpuscular Haemoglobin (pg) | 28.55 (27.15, 30.2) | 27–32 |
| Mean Corpuscular Haemoglobin Concentration (g/dl) | 29.5 (28.25, 31.3) | 32–36 |
| Red Cell Distribution Width Coefficient (%) | 16.25 (14.45, 17.7) | 11.5–14.5 |
| Red Cell Distribution Width Standard Deviation (fl) | 56.63545 (49, 60.6) | 40–55 |
| Platelets (x $10^9$/L) | 206.5 (149.5, 336) | 140–440 |
| Neutrophil Count (x $10^9$/L) | 7.145 (3.93, 10.75) | 2–7.5 |
| Neutrophil Percentage (%) | 78.55 (60.55, 85.55) | 51–76 |
| Lymphocyte Count (x $10^9$/L) | 1.105 (0.69, 2.085) | 1–4 |
| Lymphocyte Percentage (%) | 13.1 (6.5, 23.75) | 20–40 |
| Eosinophil Count (x $10^9$/L) | 0.01 (0, 0.03) | 0–0.45 |
| Eosinophil Percentage (%) | 0.1 (0, 0.4) | 0–5 |
| Monocyte Count (x $10^9$/L) | 0.58 (0.17, 0.85) | 0.18–0.8 |
| Monocyte Percentage (%) | 5.3 (2.35, 8.4) | 5–8 |
| Basophil Count (x $10^9$/L) | 0.025 (0.01, 0.04) | 0–0.2 |
| Basophil Percentage (%) | 0.3 (0.13, 0.6) | 0–0.2 |
| Immature Granulocyte Count (x $10^9$/L) | 0.31 (0.1, 0.62) | 0–0.03 |
| Immature Granulocyte Percentage (%) | 2.7 (1.2, 6.4) | 0–0.5 |
| Neutrophil to Lymphocyte Ratio | 6.02(2.5, 12.2) | - |

eosinophil counts ($r_s$ = 0.40; $p$ = 0.014), neutrophil counts ($r_s$ = 0.40; $p$ = 0.013) and immature granulocyte counts ($r_s$ = 0.55; $p$ = 0.001). Conversely, basophils were negatively correlated with monocyte counts ($r_s$ = - 0.34, 0.034) and lymphocyte counts ($r_s$ = - 0.41; $p$ = 0.011). Immature granulocytes also displayed significant correlations with neutrophils, they exhibited a positive correlation with neutrophil counts ($r_s$ = 0.38; $p$ = 0.033).

**Table 3. Summary of lymphocyte profiles of hospitalised COVID-19 patients.**

| Variable | COVID-19 Patients Median (Q1, Q3) | Reference Range |
|---|---|---|
| CD3[+] Lymphocyte Percentage (%) | 51 (37, 70) | 55–84 |
| CD3[+] Lymphocyte Count (cells/μl) | 666 (269, 1340) | 690–2540 |
| CD8[+] CD4[-] Lymphocyte Percentage (%) | 27 (17, 35) | 13–41 |
| CD8[+] CD4[-] Lymphocyte Count (cells/μl) | 273 (144, 571) | 190–1140 |
| CD4[+] CD8[-] Lymphocyte Percentage (%) | 12 (2, 28.98) | 31–60 |
| CD4[+] CD8[-] Lymphocyte Count (cells/μl) | 132 (29, 388) | 410–1590 |
| CD16[+]/CD56[+]/CD16[+]CD56[+] Lymphocyte Percentage (%) | 5 (3, 15) | 5–27 |
| CD16[+]/CD56[+]/CD16[+]CD56[+] Lymphocyte Count (cells/μl) | 79.5 (42, 206) | 90–590 |
| CD19[+] Lymphocyte Percentage (%) | 17.5 (6, 32) | 6–25 |
| CD19[+] Lymphocyte Count (cells/μl) | 326 (63, 572) | 90–660 |

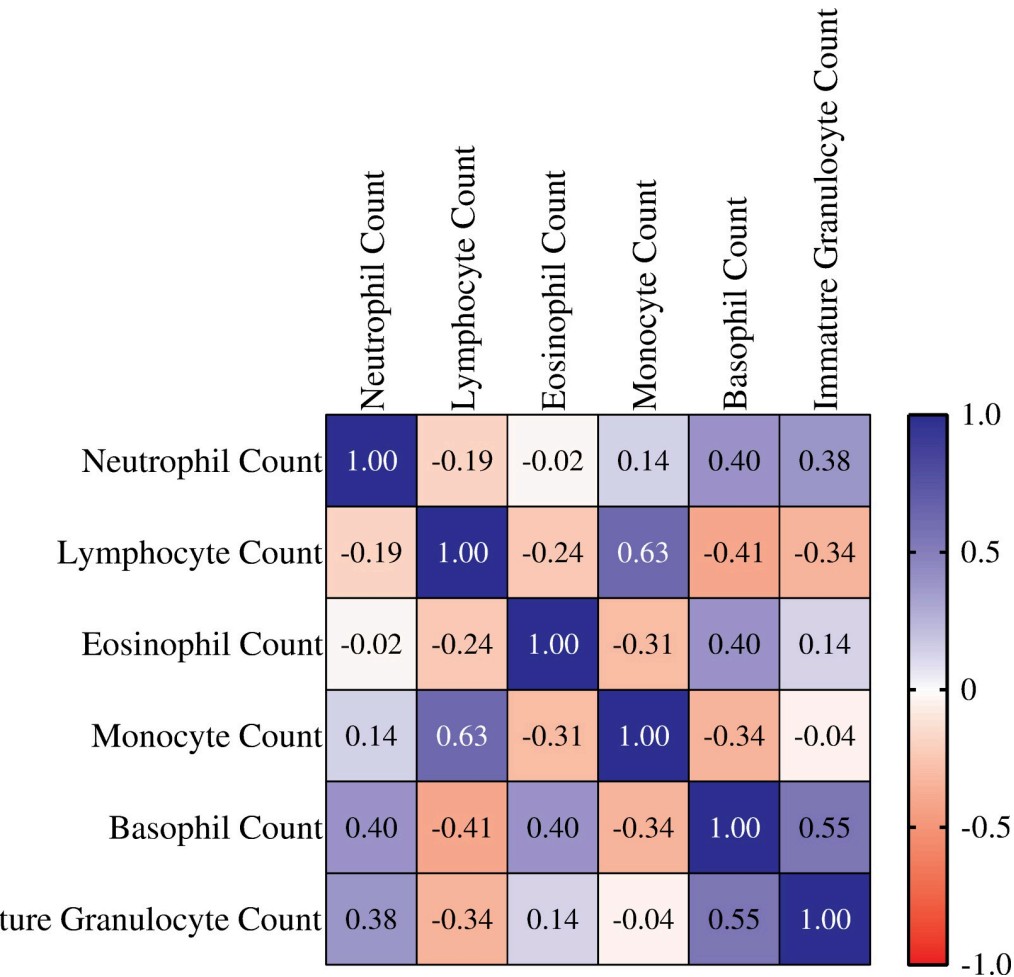

**Fig 1. Spearman's correlation matrix of cell populations in COVID-19 patients.** The correlation matrix was obtained by analysing the relationship between the white blood cell differential counts of COVID-19 patients. The correlations were determined by $r_s$ values in the matrix. Each cell was colour coded according to a heat-map that depicts the measure of correlation, blue for a positive correlation and red for a negative correlation.

## Qualitative analysis of C-reactive protein

Qualitative analysis of CRP expression was carried out using serum samples. The results showed that 27 patients (85.17%) were CRP positive, whilst 5 patients (14.29%) were CRP negative (Table 5). A Fischer's exact test was carried out to determine an association between CRP

**Table 4. Probability values of Spearman's correlation between leucocytes.**

|  | Neutrophil Count | Lymphocyte Count | Eosinophil Count | Monocyte Count | Basophil Count | Immature Granulocyte Count |
|---|---|---|---|---|---|---|
| Neutrophil Count |  | 0.248 | 0.886 | 0.377 | 0.013 | 0.033 |
| Lymphocyte Count | 0.248 |  | 0.151 | < 0.0001 | 0.011 | 0.059 |
| Eosinophil Count | 0.886 | 0.151 |  | 0.061 | 0.014 | 0.455 |
| Monocyte Count | 0.377 | < 0.0001 | 0.061 |  | 0.034 | 0.825 |
| Basophil Count | 0.013 | 0.011 | 0.014 | 0.034 |  | 0.001 |
| Immature Granulocyte Count | 0.033 | 0.059 | 0.455 | 0.825 | 0.001 |  |

**Table 5. Distribution of C-reactive positive patients with respect to neutrophil lymphocyte ratio.**

| Laboratory Test | COVID-19 patients with low NLR | COVID-19 patients with high NLR | p |
|---|---|---|---|
| C-reactive Protein: | | | |
| Positive, n (%) | 12 (70.59) | 15 (100) | 0.046 |
| Negative, n (%) | 5 (29.41) | 0 (0) | |

expression and the patients' NLR scores. CRP expression was observed to be significantly associated with NLR scores of the recruited patients ($p = 0.046$) (Table 5). While 70.59% of the patients in the low NLR category were CRP positive, 100% of the patients within the high NLR category were CRP positive.

## Haematological features of COVID-19 patients stratified by NLR

After NLR stratification, it was observed that there was no significant statistical difference in the distribution of the haematological features and NLR was not associated with the demographic features of the patients (S8 Table). Notably, the high NLR group had decreased mean corpuscular volume values and mean corpuscular haemoglobin values, whilst the platelets counts increased for the same group.

## Distribution of granulocytes and monocytes in COVID-19 patients stratified by NLR

The population counts of immature granulocytes, eosinophils, monocytes and basophils were presented in column scatter plots (Fig 2). When the Mann-Whitney $U$ test was used to compare the distributions, basophil counts and immature granulocyte counts were noted to be significantly higher for patients within the high NLR group ($p = 0.0391$; $p = 0.0165$) (Fig 2A and 2C). However, the basophil counts were also noted to be within the upper and lower limit (UL and LL) of the normal ranges. Monocyte counts were observed to be significantly higher for the low NLR groups ($p = 0.0438$) (Fig 2D) and the distribution was spread above the UL of the normal range. Eosinophils were observed to have no significant statistical difference in distribution of the 2 groups.

## Analysis of cytokines expressed by COVID-19 patients

After the patients were stratified by their NLR scores, a Kruskal-Wallis test was used to determine significant differences in cytokine expression. Serum IL-2 and IL-4 were not detected in the samples of the patients (Fig 3B and 3C). Contrastingly, there was a prevalent expression of IL-10 and IL-6 amongst the collective group of recruited patients. Patients within the high NLR group were noted to have significantly higher expression of IL-10 ($p < 0.001$) than the control group and low NLR patients also exhibited significantly higher expression of IL-10 ($p < 0.01$) than the control group. However, there was no significant difference in IL-10 expression between the high and low NLR groups (Fig 3E). Similarly, both the high and low NLR group exhibited a significantly higher expression of IL-6 ($p < 0.001$) compared to the control. No significant difference in IL-6 expression was observed between the low and high NLR groups (Fig 3D). Although high NLR patients were noted to express significantly higher levels of IFN-γ in serum compared to the healthy controls ($p < 0.01$), there was no statistical difference in expression between the high and low NLR groups (Fig 3A). TNF-α was mainly expressed by patients within the low NLR group and a strong statistical difference in TNF-α expression compared to the control group ($p < 0.01$) was observed. There was no significant

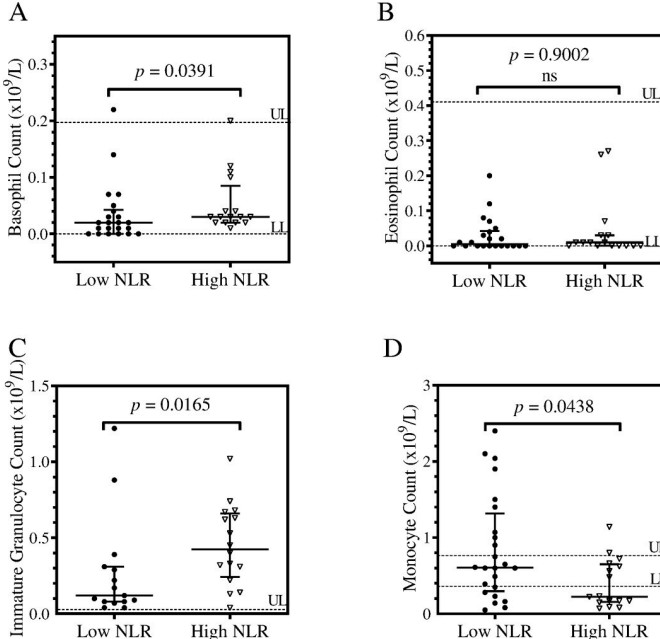

**Fig 2. Column scatter plots of leucocyte subset counts of COVID-19 patients.** The patients were stratified by their NLR scores. Each graph shows the distribution of a subset of leucocytes, which are basophils (A), eosinophils (B), immature granulocytes (C) and monocyte (D). Each column represents an NLR group, showing the counts observed in each patient, the median and interquartile ranges. An analysis of the distribution was made using the Mann-Whitney *U* test at 95% significance interval. The upper limit (UL) and lower limit (LL) demarcate the normal ranges.

statistical difference in TNF-α in expression between the low NLR and high NLR groups (Fig 3G). The most significant finding of this study was observed when significantly higher incidences in IL-17A expression were noted within the high NLR group. Patients with the high NLR group were noted to express significantly higher levels of IL-17A compared to low NLR patients ($p < 0.01$) and the control group ($p < 0.001$) (Fig 3F).

## Distribution of lymphocyte subsets in COVID-19 patients stratified by neutrophilia status

Neutrophils exert versatile functions in the immune system, thus, a neutrophilia stratification was carried out to investigate how neutrophils may contribute to lymphocyte subset expansion. However, there no significant statistical differences in distribution that were noted (S1 Fig). Notably, neutrophilia patients had higher CD4+ lymphocyte percentages, CD16+, CD56+ and CD16+CD56+ lymphocyte percentages and CD4+/CD8+ lymphocyte ratios. Non-neutrophilia patients exhibited higher CD3+ lymphocyte counts and CD19+ lymphocyte counts (S1 Fig).

## Correlation between lymphocyte subset and monocytes

A Spearman's correlation test between monocyte percentages and the lymphocyte subset counts was done to investigate the correlation between monocytes and lymphocytes. Graphs show a strong and statistically significant correlation between monocyte percentages and all lymphocyte subset counts, except CD16+CD56+ lymphocytes (Fig 4). CD19+ lymphocytes exhibited the strongest correlation ($r_s = 0.747$; $p < 0.0001$), followed by CD3+ lymphocytes ($r_s = 0.636$; $p < 0.0001$), then CD8+ lymphocytes ($r_s = 0.560$; $p = 0.00011$) and finally, CD4+ lymphocytes ($r_s = 0.480$; $p = 0.001$).

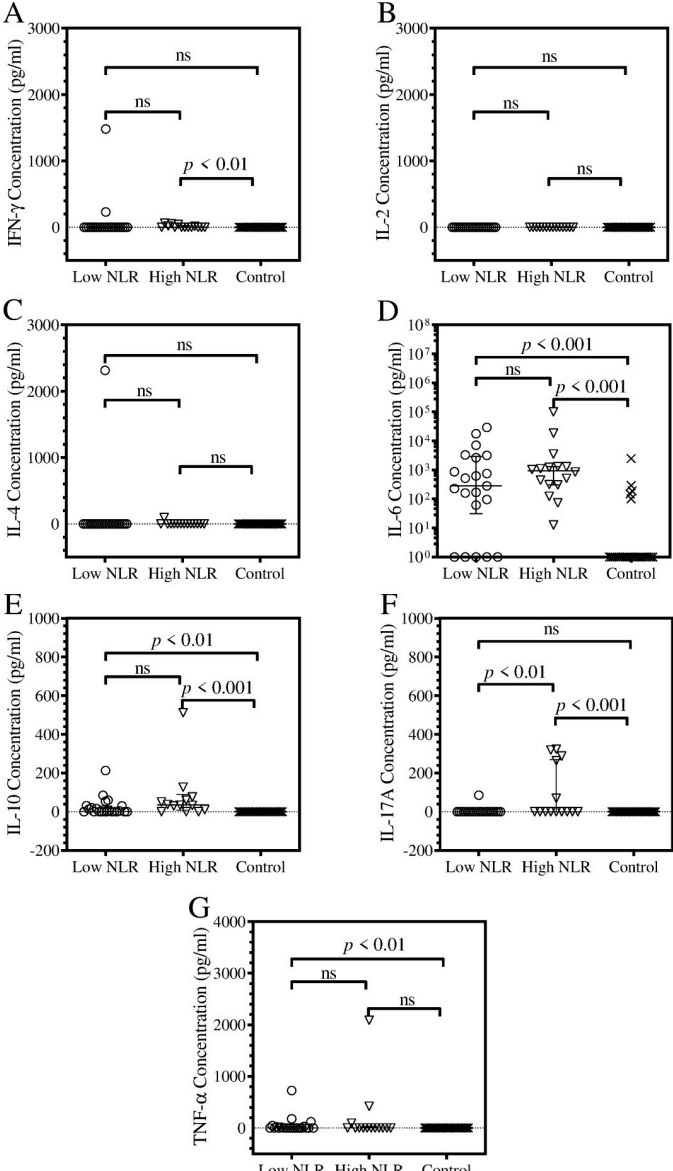

**Fig 3. Column scatter plots of cytokines expressed by COVID-19 patients.** The patients were stratified by their NLR scores. Each graph shows the concentration of the cytokines, IFN-γ (A), IL-2 (B), IL-4 (C), IL-6 (D), IL-10 (E), IL-17A (F) and TNF-α (G). The scatter plot of each NLR group and the healthy controls shows the concentration of the cytokines for each patient, the median and interquartile ranges. An analysis of the distributions was performed using the Kruskal-Wallis test at 95% significance interval, with Dunn's multiple comparison post-test.

## Discussion

The spread of COVID-19 is difficult to control and the disease continues to claim lives. The progression of the disease has been driven by successive waves of infections regionally [4]. Consequently, the SARS-CoV-2 virus brought an unprecedented threat globally. According to preceding reports, COVID-19 is markedly heterogeneous and at least 14% of all infected individuals may exhibit severe symptoms [7, 16, 20]. More importantly, severe COVID-19 cases are distinctively characterised by neutrophilia and lymphopenia, which may lead to impaired viral clearance and poor outcomes [10, 15, 17]. Guided by initial studies, this study was set to

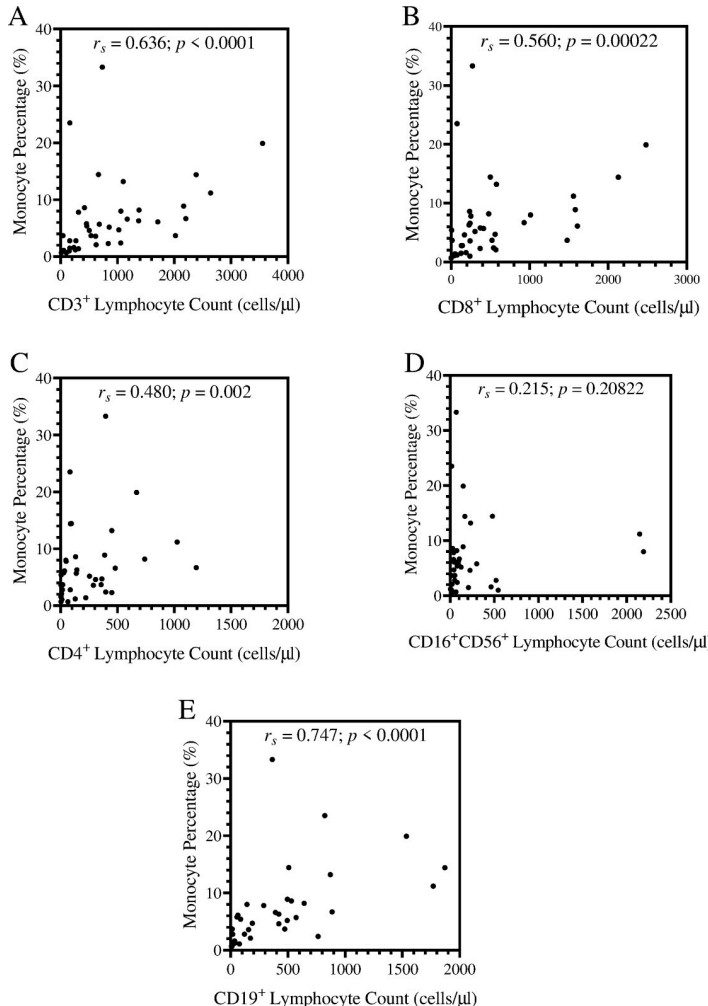

**Fig 4. Scatter plot showing the correlation between different lymphocyte subsets and monocyte percentage.** Each graph shows dot plots of all monocyte percentages and lymphocyte subset counts. The graphs show CD3+ lymphocytes (A), CD8+ lymphocytes (B), CD4+ lymphocytes (C), CD16+CD56+ lymphocytes (D) and CD19+ lymphocytes (E). The correlations were analysed by a two tailed Spearman's correlation test at 95% significance interval, where $p < 0.05$ was considered significant and $p < 0.01$; $p < 0.001$ were used to determine the magnitude of significance.

investigate how neutrophilia and lymphopenia are linked with the cellular responses and cytokine responses in COVID-19 patients. Results from this study provide a unique perspective on the immunopathology of COVID-19, with a particular focus on neutrophilia and lymphopenia.

A total of 43 COVID-19 patients were recruited from Parirenyatwa General Hospital, 69.77% of which were male and 30.23% were female. Although there are limited reports that describe the demographic characteristics of hospitalised COVID-19 patients in sub-Saharan Africa, the demographic characteristics from this study were consistent with an earlier retrospective study [21]. The study was conducted in in the Democratic Republic of the Congo (DRC) and the median age from the study was noted to be 46 years and of the 766 COVID-19 patients, 65.6% were male [21]. Comparably, the median age and the interquartile range were noted to be 48.72 (40, 60) years and 69.77% of the patients were male. These demographic features are indicative of the population that is at risk of hospitalisation. However, a wider retrospective study may help ascertain these insights.

Analysis of complete blood counts of COVID-19 patients revealed that the percentages of leucocyte subsets were skewed towards higher neutrophil and immature granulocyte percentages. Whilst neutrophils and immature granulocytes had higher percentages, CD4[+] T-lymphocytes populations and percentages of most patients were observed to be low. Neutrophil, immature granulocyte and CD4[+] T-lymphocyte populations may be heavily implicated in the immunopathology of COVID-19. A meta-analysis study observed that the levels of neutrophils increase while the levels lymphocyte decrease [22]. Such an imbalance entails prolonged innate immune responses, which may promote release of cytotoxic granules at sites of infection, NETosis and enhanced coagulation [23]. Prolonged innate immune responses may overpower lymphocytes that dampen inflammatory responses [23]. Furthermore, since CD4[+] T-lymphocytes were observed to be low in most patients, it is important to investigate human immunodeficiency virus (HIV) and SARS-CoV-2 coinfections.

A NLR stratification was carried out to provide a unique perspective on the inflammatory events in COVID-19 patients. The NLR index has been demonstrated to predict hyper-inflammatory status in patients [24–26]. Hence, the rationale behind the NLR stratification was to reflect unresolved inflammatory responses and predict the different cytokines and cell types that confer these responses. An advantage of using this index is that it can then be used to discern the inflammatory events taking place in a low resource healthcare setting [27]. From the results, CRP expression was noted to be indicative of severe disease, since 100% of patients with a higher NLR score were CRP positive. However, CRP titres could have improved this analysis. Additionally, there were no observed significant statistical differences between the haematological parameters of patients in the high NLR group and patients the low NLR group. Patients in the high NLR group had higher platelet counts compared to low NLR group, which indicates a risk of accelerated clot formation. Accelerated clot formation may occur as a consequence of platelet-neutrophil complexation, and promote a pro-thrombic environment, hyper-inflammation and prolonged neutrophil survival [23]. Hence, there might be a need to monitor the NLR and platelet counts of COVID-19 patients.

Whilst circulating basophils and immature granulocytes were significantly higher in counts, monocytes were significantly lower for patients in the high NLR group. According to these observations, immature granulocytes and basophils could be heavily involved in inflammatory responses whilst monocytes may have a protective role. These results were also emphasized by the Spearman's correlation analysis of leucocytes. Elevated immature granulocytes reflect prolonged innate immune response in the high NLR group, which may be due to prolonged stimulation of the bone marrow [28]. These findings also point to cases of emergency myelopoiesis in patients with a high NLR. Together with neutrophils, immature granulocytes may aggravate inflammation and may eventually lead to ARDS [28]. Basophils have been neglected as a leucocyte subset possibly due to their minority and redundancy in roles with mast cells [29]. Nonetheless, basophils interact with other cells making use of basophil derived factors that can contribute to inflammatory responses and they have been noted to augment T-helper 17 responses and IL-17 production [29]. Thus, it may be important to revisit the role of basophils in inflammation, especially as a result of viral infections. Monocytes may be implicated in the hyper-inflammatory processes associated with COVID-19 infections [13]. Monocyte counts were lower in the high NLR group, which was unexpected. Thus, it may be crucial to determine the dominant immuno-phenotypes of circulating monocytes for both NLR groups.

Severe cases of COVID-19 are characterised by a cytokine storm, which was earlier reported to be reminiscent of a macrophage activation syndrome [5, 30, 31]. Macrophage activation syndrome is typified by elevated levels of IFN-γ [15]. The concentration of IFN-γ in the serum of COVID-19 patients was relatively lower. Results of this study were concordant with another study that examined T-helper 1 and T-helper 2 cytokines. The study reported a prevalent

expression of IL-6 and IL-10 and lower expression of TNF-α, IL-2, IL-4 and IFN-γ [11]. More importantly, high levels of IL-17A expression and relatively lower IFN-γ gamma expression suggest skewed CD4$^+$ T-lymphocytes polarisation towards a pro-inflammatory T-helper 17 subset in the high NLR group. It appears as if the T-helper 17 responses in the high NLR group results in inflammation and unabated IL-6 and IL-10 expression. Neutrophils and IL-6 are mediators of T-helper 17 polarisation of naïve CD4$^+$ T-lymphocytes [32]. Since severe COVID-19 is hallmarked by neutrophilia, stratification using the NLR index provided a significant perspective on the immunopathology of the disease. It is possible that as the severity of the disease progresses, prolonged and excessive neutrophil recruitment, as well as IL-6 expression among other factors, may provide a milieu that promotes T-helper 17 and IL-17 production. Although, the neutrophilia stratification did not reveal significant statistical differences in the distribution of lymphocyte subsets, patients with neutrophilia had elevated CD4$^+$ T-lymphocyte percentages and CD4$^+$/CD8$^+$ T-lymphocyte ratio. These results support earlier findings that directly illustrated how neutrophils mediate the T-helper 17 promotion in COVID-19 patients [33]. Therefore, inhibiting the pivotal events that promote T-helper 17 pro-inflammatory may provide a key therapeutic strategy for treating severe cases of COVID-19.

The Spearman's correlation matrix highlighted a strong and significant correlation between circulating lymphocytes and monocytes. This correlation was further investigated by analysing the correlation between lymphocyte subset counts and monocyte percentages. Correlation between lymphocyte subsets and monocytes aimed revealing the overall landscape of lymphocyte activation and expansion. CD4$^+$ T-lymphocytes had the weakest correlation among the lymphocyte subsets, excluding natural killer (CD16$^+$CD56$^+$) lymphocytes. CD4$^+$ T-lymphocytes rely largely on MHC II antigen presentation, and monocytes develop into antigen presenting cells that present antigens via the MHC II molecules [34]. Thus, indicating a dysfunctional antigen presentation.

This study was limited by a number of challenges and facets. First, participating patients were recruited and their blood sampled within an identical time period, which was on admission. Since there was no subsequent sampling, only baseline observations were made. There is a possibility that some of the responses can only be observed within a time frame that was overlooked in this regard, thus making it difficult to draw a more comprehensive conclusion. While this study only took into consideration hospitalised patients, studying the entire disease spectrum including those that were not hospitalised may help derive some of the factors that confer protection. Another limitation is that the only peripheral blood was drawn from the patients. Peripheral blood mostly provides a reliable perspective on the key biomarkers and cellular populations [35]. Since COVID-19 is a respiratory disease that can manifest in different ways, it is possible that there are some key events that could be occurring at sites of infections. Sampling bronchoalveolar fluid, for example, and relating it to the NLR index may provide a holistic analysis of the mediators that contribute to the excessive neutrophil recruitment and potentially provide a rationale for drug design and therapeutic strategies.

## Conclusion

Conclusively, COVID-19 patients exhibited white blood cell percentages that were skewed in favour of increased neutrophil and immature granulocyte percentages. When the patients were stratified by their NLR scores, patients categorised in the high NLR group were noted to have elevated levels of immature granulocytes and basophils, and lower monocyte counts compared to those in the low NLR group. The granulocytic and neutrophilic environment in severe COVID-19 patients was shown to promote a typical T-helper 17 response. The response was significantly marked by skewed CD4$^+$ T-lymphocyte expansion demonstrated by relatively

high IL-17A expression and low IFN-γ expression. Consequently, inflammation proceeded unabatedly pronounced by IL-10 and IL-6 expression for patients within the high NLR Group. Therefore, these results support that excessive neutrophil recruitment in COVID-19 patients may drive a T-helper 17 response as previously demonstrated by an earlier study [33]. Based on these findings, monoclonal antibodies targeting the IL-17 and IL-6 signalling pathways, targeting neutrophil activity, systemic corticosteroids and broad activity immunomodulatory drugs can help dampen the hyper-inflammatory events that are driven by neutrophils in COVID-19 patients.

## Supporting information

**S1 Table. Demographic information of COVID-19 patients.**
(DOCX)

**S2 Table. Data of haematological parameters of COVID-19 patients.**
(DOCX)

**S3 Table. White blood cell differential data of COVID-19 patients.**
(DOCX)

**S4 Table. Data of lymphocyte subset populations of COVID-19 patients.**
(DOCX)

**S5 Table. Data of CRP expression by COVID-19 patients.**
(DOCX)

**S6 Table. Cytokine concentrations in COVID-19 patients' serum.**
(DOCX)

**S7 Table. Cytokine concentrations in control subject's serum.**
(DOCX)

**S8 Table. Demographic and haematological features of COVID-19 patients with respect to neutrophil lymphocyte ratio (NLR).**
(DOCX)

**S1 Fig. Column scatter plots of lymphocyte subsets of COVID-19 patients.**
(PDF)

## Acknowledgments

The authors would like to acknowledge Gamma Allergy and Immunology Laboratory staff, Haematology Centre, TIBA Zimbabwe and the University of Zimbabwe for their contribution and efforts to the study.

## Author Contributions

**Conceptualization:** Elopy Nimele Sibanda, Takafira Mduluza.

**Data curation:** Tanaka Arthur Choto.

**Formal analysis:** Tanaka Arthur Choto.

**Funding acquisition:** Elopy Nimele Sibanda, Takafira Mduluza.

**Investigation:** Tanaka Arthur Choto, Ian Makupe, Andrew Zolani Cakana, Elopy Nimele Sibanda, Takafira Mduluza.

**Methodology:** Andrew Zolani Cakana, Elopy Nimele Sibanda, Takafira Mduluza.

**Project administration:** Elopy Nimele Sibanda, Takafira Mduluza.

**Resources:** Andrew Zolani Cakana, Elopy Nimele Sibanda, Takafira Mduluza.

**Supervision:** Andrew Zolani Cakana, Elopy Nimele Sibanda, Takafira Mduluza.

**Validation:** Ian Makupe.

**Visualization:** Tanaka Arthur Choto.

**Writing – original draft:** Tanaka Arthur Choto.

**Writing – review & editing:** Tanaka Arthur Choto, Elopy Nimele Sibanda, Takafira Mduluza.

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
