## [Decision Letter · Decision Letter 0]

13 Apr 2022

PONE-D-22-00109Excessive Neutrophil Recruitment Promotes Typical T-helper 17 Responses in Coronavirus Disease 2019 PatientsPLOS ONE

Dear Dr. Choto,

Thank you for submitting your manuscript to PLOS ONE. After careful consideration, we feel that it has merit but does not fully meet PLOS ONE’s publication criteria as it currently stands. Therefore, we invite you to submit a revised version of the manuscript that addresses the specific points raised by the external reviewer in a point-by-point reply.

We look forward to receiving your revised manuscript.

Kind regards,

Paul Proost, Ph.D.

Academic Editor

PLOS ONE

Journal Requirements:

The authors declare that the research was conducted in the absence of any commercial or financial relationships that could be construed as a potential conflict of interest.

Reviewers' comments:

Reviewer's Responses to Questions

**Comments to the Author**

1. Is the manuscript technically sound, and do the data support the conclusions?

Reviewer #1: Yes

2. Has the statistical analysis been performed appropriately and rigorously? 

Reviewer #1: Yes

3. Have the authors made all data underlying the findings in their manuscript fully available?

Reviewer #1: Yes

4. Is the manuscript presented in an intelligible fashion and written in standard English?

Reviewer #1: Yes

5. Review Comments to the Author

Reviewer #1: In this manuscript, the authors provide a descriptive data-set analyzing leukocytes and some cytokines in a small cohort of hospitalized COVID-19 patients. The study is confirming results from several other studies showing lymphopenia and elevated (immature) neutrophils (and so elevated neutrophil-to-lymphocyte ratios) in blood from COVID-19 patients and its association with COVID-19 severity. Moreover, the authors discovered elevation of some cytokines in serum of COVID-19 patients, confirming the cytokine storm occurring in these patients. As such, the study is not really novel, but a confirmation of the typical findings seen in severe COVID-19. Here are my remarks:

- In the abstract, the authors mentioned that “the results show that a systemic neutrophilic environment may preferentially skew CD4+ lymphocytes towards T-helper 17 and IL-17A promotion”. Although this might be true, this is NOT directly shown by the results. The authors could only speculate this based on the elevated IL-17A levels. Other studies (e.g. Parackova et al., J Leukoc Biol, 2021) really show that COVID-19 neutrophils promote the induction of Th17 cells. The authors should include this in their discussion and rephrase their abstract and conclusion.

- 43 hospitalized COVID-19 patients were included in the study. Can the authors provide additional information about these patients? Were these patients admitted to intensive care units (ICUs), did they get respiratory support? If and how were these patients treated for their COVID-19 disease? Moreover, in the different experiments, not always the same number of patients is included. The authors should provide the reason for excluding some patients in some experiments. Moreover, authors should include how many healthy controls were included in the study in the methods section.

- Blood samples were collected during the first 8 days of hospitalization. Can the authors do a kinetics analysis with the data? As shown by Metzemaekers et al. (Clin Transl Immunology, 2021), neutrophil function and cytokine responses could be significantly different shortly after admission to ICU compared to one week stay in ICU.

- In the data analysis, the authors chose for a NLR value of 7.5 to stratify patients in two groups. What is the rationale to choose 7.5 as a cut-off?

- Discussion section:

o Line 452-454: Elevated immature granulocytes are linked to prolonged innate immune responses and prolonged stimulation of the bone marrow. Emergency myelopoiesis should also be indicated.

o Line 461-462: It is shown in the results that monocytes are lowered, but nowhere it is shown that they have a protective role.

o Line 497-500 and line 529: Too much speculation which should be removed.

o Line 525: downregulated IFN-y expression: downregulated compared to what?

- Revision of the tables + figures + references should be made:

o In some tables median (Q1-Q3) should be added.

o The NLR value could be added to table 2.

o Table 6 could be moved to the Supplementary material, if necessary, as no significant differences are shown. Moreover, if the differences are not significant, authors should only talk about a trend and should not stress differences too much when discussing results. Same for Figure 4.

o Table 7 could be combined with Table 5 as Table 5 alone does not contain a lot of useful information. Table 7 shows the relation of CRP with severity, which is much more useful.

o The order of the panels in Figure 3 should be changed so that they are in a logical order corresponding to the text. In panel 3B IL-4 instead of IL-2 should be stated.

o In the figure legends, authors should remove asterisks for significance if it is not visible in the figures.

o References should be mentioned in chronological order throughout the text.

- English writing could be improved + authors should carefully check for typo’s e.g. In the text for Figure 1 there is no significant correlation between immature granulocyte count and lymphocyte count (p = 0.059) as it is now incorrectly mentioned in the results. Moreover, the rs-values and p values are not corresponding to the ones found in Table 4.

6. PLOS authors have the option to publish the peer review history of their article (what does this mean?). If published, this will include your full peer review and any attached files.

Reviewer #1: **Yes: **Seppe Cambier

---

## [Author Response · Author response to Decision Letter 0]

23 Jun 2022

Reviewer: Please ensure that your manuscript meets PLOS ONE's style requirements, including those for file naming. The PLOS ONE style templates can be found at https://journals.plos.org/plosone/s/file?id=wjVg/PLOSOne_formatting_sample_main_body.pdf and https://journals.plos.org/plosone/s/file?id=ba62/PLOSOne_formatting_sample_title_authors_affiliations.pdf

Response: The manuscript and title authors’ affiliations have been revised to meet style requirements. 

Reviewer: Please provide additional details regarding participant consent. In the ethics statement in the Methods and online submission information, please ensure that you have specified what type you obtained (for instance, written or verbal, and if verbal, how it was documented and witnessed). If your study included minors, state whether you obtained consent from parents or guardians. If the need for consent was waived by the ethics committee, please include this information.

Response: This study was conducted after the protocol was reviewed and approved by the Medical Research Council of Zimbabwe approval MRCZ/A/2602. Permission to conduct the study was also obtained from the Joint Research Ethics Committee of Parirenyatwa Group of Hospitals and the University of Zimbabwe College of Health Sciences. After having thorough discussion on the procedures and purpose of the study and before commencement of data collection, written consent was obtained from the participants. Patients requiring ICU management were not included in the study. These could not give informed consent.

An additional ethics statement section has been included in the methods section mentioning how the consent was obtained. 

Reviewer: We note that the grant information you provided in the ‘Funding Information’ and ‘Financial Disclosure’ sections do not match.

Response: This has been corrected online to match the information provided in the manuscript.

Reviewer: The authors declare that the research was conducted in the absence of any commercial or financial relationships that could be construed as a potential conflict of interest. Please complete your Competing Interests on the online submission form to state any Competing Interests. If you have no competing interests, please state "The authors have declared that no competing interests exist.” as detailed online in our guide for authors at http://journals.plos.org/plosone/s/submit-now this information should be included in your cover letter; we will change the online submission form on your behalf.

Response: The information has been provided on the online submission. 

Reviewer: We note that you have stated that you will provide repository information for your data at acceptance. Should your manuscript be accepted for publication, we will hold it until you provide the relevant accession numbers or DOIs necessary to access your data. If you wish to make changes to your Data Availability statement, please describe these changes in your cover letter and we will update your Data Availability statement to reflect the information you provide.

Response: All the data has been provided as Supplementary Information files. 

Reviewer: Please include captions for your Supporting Information files at the end of your manuscript, and update any in-text citations to match accordingly. Please see our Supporting Information guidelines for more information: http://journals.plos.org/plosone/s/supporting-information.

Response: Supporting Information file captions have been added within the manuscript file (at the end). The in-text citations have been updated accordingly. 

Reviewer: Please review your reference list to ensure that it is complete and correct. If you have cited papers that have been retracted, please include the rationale for doing so in the manuscript text, or remove these references and replace them with relevant current references. Any changes to the reference list should be mentioned in the rebuttal letter that accompanies your revised manuscript. If you need to cite a retracted article, indicate the article’s retracted status in the References list and also include a citation and full reference for the retraction notice

Response: The reference list has been reviewed. It is complete and correct. All papers that were cited are valid, and none of them have been retracted. Reference 33 has been added, to reference the findings supported by this manuscript. 

Reviewer: In the abstract, the authors mentioned that “the results show that a systemic neutrophilic environment may preferentially skew CD4+ lymphocytes towards T-helper 17 and IL-17A promotion”. Although this might be true, this is NOT directly shown by the results. The authors could only speculate this based on the elevated IL-17A levels. Other studies (e.g. Parackova et al., J Leukoc Biol, 2021) really show that COVID-19 neutrophils promote the induction of Th17 cells. The authors should include this in their discussion and rephrase their abstract and conclusion.

Response: The sentence in the abstract has been changed to - “Results from this study suggest that a systemic neutrophilic environment may preferentially skew CD4+ lymphocytes towards T-helper 17 and IL-17A promotion, thus, aggravating inflammation.” 

Earlier finding by Parackova et al., 2021, has been cited in the discussion - “Additionally, these results support earlier findings that directly illustrated how neutrophils mediate the T-helper 17 promotion in COVID-19 patients [33].”

The sentence in the conclusion has been revised to - “Therefore, these results suggest that excessive neutrophil recruitment in COVID-19 patients may drive a T-helper 17 response as previously demonstrated by an earlier study [33]”

Reviewer: 43 hospitalized COVID-19 patients were included in the study. Can the authors provide additional information about these patients? Were these patients admitted to intensive care units (ICUs), did they get respiratory support? If and how were these patients treated for their COVID-19 disease? Moreover, in the different experiments, not always the same number of patients is included. The authors should provide the reason for excluding some patients in some experiments. Moreover, authors should include how many healthy controls were included in the study in the methods section.

Response: The patients were admitted according to the hospital protocols. The hospital designates an isolation section for COVID-19 patients, where the patients were recruited for the study. Hence, ICU patients were not recruited into the study. Information regarding treatment and respiratory support was not provided. 

For some patients, the volume of blood that was drawn was not sufficient enough for all the possible tests. As a result, a few of them had missing results for tests like C-reactive protein test. 

The number of controls has been mentioned in the methods section as advised. 

Reviewer: Blood samples were collected during the first 8 days of hospitalization. Can the authors do a kinetics analysis with the data? As shown by Metzemaekers et al. (Clin Transl Immunology, 2021), neutrophil function and cytokine responses could be significantly different shortly after admission to ICU compared to one week stay in ICU.

Response: The blood samples were collected only once, therefore no follow-up data is available to perform a kinetics analysis as suggested. However, this limitation was emphasised in the discussion - “First, participating patients were recruited and sampled within an identical time period, which was during the first 8 days of admission. Since there was no subsequent sampling which was carried out, only baseline observations were made. There is a possibility that some of the responses can only be observed within a time frame that was overlooked in this regard, thus making it difficult to draw a more comprehensive conclusion.” Please note that it is emphasized that no ICU patients were included in the study due to complication of their conditions and difficult to obtain consent.

Reviewer: In the data analysis, the authors chose for a NLR value of 7.5 to stratify patients in two groups. What is the rationale to choose 7.5 as a cut-off?

Response: The rationale behind 7.5 as the cut-off has been added to the methods section - “The rationale behind the 7.5 as the cut-off value was to target neutrophilia patients (neutrophil count > 7.5 cells x 109/L) and/or lymphopenia patients (lymphocyte count > 7.5 cells x 109/L) as severe COVID-19 patients. Therefore, a ratio of 7.5 and above was deemed as a suitable cut-off value.” 

Reviewer: Discussion Line 452-454: Elevated immature granulocytes are linked to prolonged innate immune responses and prolonged stimulation of the bone marrow. Emergency myelopoiesis should also be indicated.

Response: Indicated as advised - “These results also indicate cases of emergency myelopoiesis for patients in the high NLR group.

Reviewer: Discussion Line 461-462: It is shown in the results that monocytes are lowered, but nowhere is it shown that they have a protective role.

Response: The claim is derived from speculation. Since a significant difference was observed in the distribution of monocytes, with patients in the high NLR group (patients with less severe COVID-19) having lower monocyte counts, they could be linked to a protective role that this study could reveal. Since this claim was made out of speculation, it has been revised to - “Monocytes have been shown to be lower for the high NLR group, which was not as expected. Thus, it may be crucial to determine the dominant immuno-phenotypes of circulating monocytes for both NLR groups.”

Reviewer: Discussion Line 497-500 and line 529: Too much speculation which should be removed.

Response: Removed as advised.

Reviewer: Discussion Line 525: downregulated IFN-y expression: downregulated compared to what?

Response: Downregulated has been omitted for lower – “The response was significantly marked by skewed CD4+ T-lymphocyte expansion demonstrated by relatively high IL-17A expression and low IFN-γ expression.”

Reviewer: In some tables median (Q1-Q3) should be added.

Response: Median (Q1, Q3) are now indicated in all tables. 

Reviewer: The NLR value could be added to table 2.

Response: Neutrophil to lymphocyte ratio (NLR) summary has been added to Table 2.

Reviewer: Table 6 could be moved to the Supplementary material, if necessary, as no significant differences are shown. Moreover, if the differences are not significant, authors should only talk about a trend and should not stress differences too much when discussing results. Same for Figure 4.

Response: Table 6 and Figure 4 were moved to the supplementary. 

Reviewer: Table 7 could be combined with Table 5 as Table 5 alone does not contain a lot of useful information. Table 7 shows the relation of CRP with severity, which is much more useful.

Response: Table 7 was combined with Table 5, to give a single set of C-reactive protein results showing an association between CRP and severity. 

Reviewer: The order of the panels in Figure 3 should be changed so that they are in a logical order corresponding to the text. In panel 3B IL-4 instead of IL-2 should be stated.

Response: The order has been revised as advised, in a logical order of the cytokines (alphabetically). 

Reviewer: In the figure legends, authors should remove asterisks for significance if it is not visible in the figures.

Response: The asterisks were removed in all the figures and their corresponding legends, since they were not visible. 

Reviewer: References should be mentioned in chronological order throughout the text.

Response: References have been reviewed and the references are mentioned in a chronological order. 

Reviewer: English writing could be improved + authors should carefully check for typo’s e.g. in the text for Figure 1 there is no significant correlation between immature granulocyte count and lymphocyte count (p = 0.059) as it is now incorrectly mentioned in the results. Moreover, the rs-values and p values are not corresponding to the ones found in Table 4.

Response: English and grammar have been reviewed as advised. The text for Figure 1 has been corrected as suggested. 

Reviewer: Please upload a copy of Figure 5 which you refer to in your text on page 17. Or if the figure is no longer to be included as part of the submission please remove all reference to it within the text.

Response: The figure referencing error has been corrected. Figure 4 is now referred to in the text as it is given.

Reviewer: Please ensure that you refer to Figure 4 in your text as, if accepted, production will need this reference to link the reader to the figure.

Response: The figure referencing error has been corrected. Figure 4 is now referred to in the text as it is given.

---

## [Decision Letter · Decision Letter 1]

4 Aug 2022

Excessive Neutrophil Recruitment Promotes Typical T-helper 17 Responses in Coronavirus Disease 2019 Patients

PONE-D-22-00109R1

Dear Dr. Choto,

We’re pleased to inform you that your manuscript has been judged scientifically suitable for publication and will be formally accepted for publication once it meets all outstanding technical requirements.

Kind regards,

George Vousden

Staff Editor

PLOS ONE

Additional Editor Comments (optional):

Reviewers' comments:

Reviewer's Responses to Questions

**Comments to the Author**

1. If the authors have adequately addressed your comments raised in a previous round of review and you feel that this manuscript is now acceptable for publication, you may indicate that here to bypass the “Comments to the Author” section, enter your conflict of interest statement in the “Confidential to Editor” section, and submit your "Accept" recommendation.

Reviewer #1: All comments have been addressed

2. Is the manuscript technically sound, and do the data support the conclusions?

Reviewer #1: (No Response)

3. Has the statistical analysis been performed appropriately and rigorously? 

Reviewer #1: (No Response)

4. Have the authors made all data underlying the findings in their manuscript fully available?

Reviewer #1: (No Response)

5. Is the manuscript presented in an intelligible fashion and written in standard English?

Reviewer #1: (No Response)

6. Review Comments to the Author

Reviewer #1: (No Response)

7. PLOS authors have the option to publish the peer review history of their article (what does this mean?). If published, this will include your full peer review and any attached files.

Reviewer #1: No

---

## [Editor Report · Acceptance letter]

9 Aug 2022

PONE-D-22-00109R1 

Excessive Neutrophil Recruitment Promotes Typical T-helper 17 Responses in Coronavirus Disease 2019 Patients 

Dear Dr. Choto:

I'm pleased to inform you that your manuscript has been deemed suitable for publication in PLOS ONE. Congratulations! Your manuscript is now with our production department. 

Kind regards, 

on behalf of

Dr. George Vousden 

Staff Editor

PLOS ONE